# Genetic Diversity and Signatures of Selection in the Roughskin Sculpin (*Trachidermus fasciatus*) Revealed by Whole Genome Sequencing

**DOI:** 10.3390/biology12111427

**Published:** 2023-11-13

**Authors:** Lize San, Zhongwei He, Yufeng Liu, Yitong Zhang, Wei Cao, Jiangong Ren, Tian Han, Bingbu Li, Guixing Wang, Yufen Wang, Jilun Hou

**Affiliations:** 1Hebei Key Laboratory of the Bohai Sea Fish Germplasm Resources Conservation and Utilization, Beidaihe Central Experiment Station, Chinese Academy of Fishery Sciences, Qinhuangdao 066100, China; 2Bohai Sea Fishery Research Center, Chinese Academy of Fishery Science, Qinhuangdao 066100, China; 3Ocean College, Hebei Agricultural University, Qinhuangdao 066009, China

**Keywords:** *Trachidermus fasciatus*, genetic diversity, population structure, selective sweeps

## Abstract

**Simple Summary:**

The roughskin sculpin (*Trachidermus fasciatus*) was previously a locally abundant species in China, but the habitat required for the survival and breeding of this species has been seriously damaged. Its resources have been almost depleted, and it has been listed as a national second-class protected animal. In recent years, the population of roughskin sculpin has recovered in the natural environment through enhancement programs and the release of juveniles. However, the effects of released roughskin sculpin on wild populations’ genetic structure and diversity remain unclear. In this study, the whole-genome resequencing of two cultured populations and one wild population showed that the two cultured populations have similar genetic diversity compared with the wild population. In addition, artificial feeding affected the formation of traits of roughskin sculpin. The finding provides data support for the resource recovery of roughskin sculpin.

**Abstract:**

The roughskin sculpin (*Trachidermus fasciatus*) is an endangered fish species in China. In recent years, artificial breeding technology has made significant progress, and the population of roughskin sculpin has recovered in the natural environment through enhancement programs and the release of juveniles. However, the effects of released roughskin sculpin on the genetic structure and diversity of wild populations remain unclear. Studies on genetic diversity analysis based on different types and numbers of molecular markers have yielded inconsistent results. In this study, we obtained 2,610,157 high-quality SNPs and 494,698 InDels through whole-genome resequencing of two farmed populations and one wild population. Both farmed populations showed consistent levels of genomic polymorphism and a slight increase in linkage compared with wild populations. The population structure of the two farmed populations was distinct from that of the wild population, but the degree of genetic differentiation was low (overall average Fst = 0.015). Selective sweep analysis showed that 523,529 genes were selected in the two farmed populations, and KEGG enrichment analysis showed that the selected genes were related to amino acid metabolism, which might be caused by artificial feeding. The findings of this study provide valuable additions to the existing genomic resources to help conserve roughskin sculpin populations.

## 1. Introduction

The roughskin sculpin (*Trachidermus fasciatus*), belonging to the Cottidae family, is a kind of small, bottom-dwelling catadromous migrating fish mainly distributed in the Gulf of Southern Japan, the west coast of the Korean Peninsula, the northeast coast of China, and the downstream of their corresponding rivers [1]. China’s roughskin sculpin was previously a locally abundant species, but due to the aggravation of environmental pollution in the surrounding seas, coupled with overfishing and the construction of dams, the habitat required for the survival and breeding of this species has been seriously damaged. Its resources have been almost depleted, and it has been listed as a national second-class protected animal [2,3]. Currently, destruction of the habitat of the roughskin sculpin in China accounts for the significant difficulties faced with respect to its research and protection. It is well established that when the population is small, species are easily affected by genetic drift, inbreeding depression, and loss of adaptive genetic variation, resulting in decreased population genetic diversity [4]. The decrease in the population’s genetic diversity affects its stability and fitness and ultimately increases the risk of extinction. Significant inroads have been achieved over the past few years in artificial breeding and mass production of seeds of roughskin sculpin, which is of great significance to protecting the species’ resources and maintaining its population size [5,6].

Elucidating the genetic diversity and genetic structure of the roughskin sculpin population is crucial for formulating conservation measures. However, there are inconsistencies in the current research on the genetic differentiation of the roughskin sculpin population. Indeed, it remains unclear whether there is a common genetic structure among Chinese roughskin sculpin populations [7]. This controversy may exist because a higher resolution than that provided by currently used molecular markers is needed, and the coverage and detection level of different molecular markers affect the reliability of the analysis of the genetic structure of the population. Over the years, random amplification of polymorphic DNA (RAPD), inter simple sequence repeat (ISSR), simple sequence repeat (SSR), mitochondrial DNA, and other molecular marker techniques have been used to study the genetic diversity and genetic differentiation levels of different populations of roughskin sculpin but yielded heterogeneous results [8,9,10]. Microsatellite markers have been used to analyze the genetic diversity of roughskin sculpin from the two geographical groups of the Yalu River and the Yellow River. It was found that the roughskin sculpin of both geographical groups had high levels genetic diversity, and the genetic diversity of the Yalu River population was slightly higher than that of the Yellow River population [11]. The microsatellite markers were used to analyze the genetic diversity of roughskin sculpin from eight geographical groups in Japan and China. The results showed that the wild roughskin sculpin had high levels of genetic diversity (Ho: 0.7~0.9) and genetic structure [12]. However, the number of microsatellite markers for a single test is limited, and microsatellites can only detect simple variations with low sensitivity, which is not conducive to other population genetic analyses, such as the study of interpopulation selection signals.

Population genomics have developed rapidly with the availability of high-throughput sequencing and the reduced costs of related technologies. Population genomics can screen thousands of genetic markers simultaneously at the genome scale, resulting in tremendous innovations in population genetics, conservation genetics, ecology, and evolutionary research [13,14,15]. For species whose genomes have been assembled, resequencing individuals of different populations based on high-throughput sequencing technology can obtain genomic information of different populations quickly and efficiently. In fish, genome-wide resequencing analyzes fish population structure and the history of population evolution by screening candidate genes associated with genomic regions and elucidating the molecular mechanisms underlying various genetic traits. In this respect, Xu et al., resequenced 33 representative carp individuals worldwide using genome-wide data, and the results showed that carp originated from two subspecies. A total of 326 candidate genes were obtained by selecting and eliminating the genes associated with carp scale deletion and red body color, which laid a foundation for revealing the molecular mechanism of trait differences in different carp (*Cyprinus carpio*) populations [16]. In a similar study of Atlantic salmon (*Salmo salar* L.), the Atlantic Ocean’s eight populations in northern and southern Norway were analyzed using genome resequencing, revealing that natural selection acting on immune-related genes led to genetic differences between populations in the Norwegian Atlantic and that the plasticity of genomes may promote differences between populations [17]. Population genetic studies conducted using molecular markers represent a necessary technical means to understand population adaptive evolution. In this respect, enrichment analysis of selected regions revealed that the expanded gene family of silver carp (*Hypophthalmichthys molitrix*) was associated with disease, immune system, and environmental adaptation [18]. Whole-genome resequencing technology can also be used to obtain the genomic information of each subpopulation of a species’ natural population, and a large amount of various information can be obtained. Through the analysis of genetic variations, various biological issues, such as population genetic structure, gene exchange, speciation mechanism, and population evolution dynamics, can be explored. This can be illustrated briefly by the evolutionary origin and domestication history of studying goldfish (*C. auratus*). With genome-wide variation information, genes associated with morphogenesis, pigmentation, behavior, immune response, infectious disease, energy metabolism, and hormone response have been successfully mapped [19].

The reference genome of roughskin sculpin has been precisely assembled, laying the groundwork for subsequent population genetic analysis and resource conservation [20]. In this study, we obtained the genome-wide SNP of roughskin sculpin through resequencing. The genetic diversity of roughskin sculpin in farmed and wild populations was analyzed based on genomic SNP, and the influence of selection on the genome of roughskin sculpin under captivity conditions was explored, guiding species conservation through artificial release.

## 2. Materials and Methods

### 2.1. Ethics

This study was performed in accordance with the Guidelines for Care and Use of Laboratory Animals provided by the Chinese Association for Laboratory Animal Sciences (No. 2011–2). The study protocols were approved by the Animal Care and Use Committee of Beidaihe Central Experiment Station, Chinese Academy of Fishery Sciences (CAFS).

### 2.2. Sample Collection and Measurement of Growth Traits

A total of 192 T. fasciatus were collected from three different geographic populations (Qinhuangdao City of Hebei province (*n* = 64, weight: 30.66 ± 10.4 g), Tianjin Municipality (*n* = 64, weight: 40.46 ± 9.84 g), and Dongying City of Shandong province (*n* = 64, weight: 30.47 ± 7.21 g), China). The detailed geographical locations of the sampling are shown in Figure 1. According to the origin of the population, they were named QHD, TJ, and SD. The SD (temperature: 12 ± 1 °C, DO: 6.5 ± 0.5 mg/L, pH: 7.5 ± 0.5, salinity: 24 ± 1‰) and TJ groups (temperature: 11 ± 1 °C, DO: 6.5 ± 0.5 mg/L, pH: 7.5 ± 0.5, salinity: 24 ± 1‰) were cultured, and QHD (temperature: 10 ± 1 °C, DO: 8.5 ± 0.5 mg/L, pH: 8.0 ± 0.5, salinity: 31 ± 1‰) was a wild population. The breeding group was taken from the farm, and the sampling time was November 2022. Wild populations were caught by bottom trawling. The fishing site was in the offshore area of the Bohai Sea, the fishing depth range was 4~6 m, the fishing route was 30 km, and the fishing time was November 2022. Their morphometric parameters were measured, including body weight, height, and length, as well as total length. Four morphological traits of (body weight (BW), body length (BL), body height (BH), and total length (TL)) were measured after anesthesia with 30 g/m^3^ MS-222. The weight was measured using and electronic balance with an accuracy of 0.1 g. The body length and height were measured with a vernier caliper, which is accurate to 0.1 cm. The measurement methods of the total length, body length, and body height of the sculpin are described as follows.

Total length: the vertical distance from the front end of the maxillary kiss to the end of the caudal fin;body length: the vertical distance from the front end of the maxillary kiss to the base of the caudal fin;body height: the vertical distance at the highest point of the fish body.

SPSS 17.0 was used to test the normality of all measured body weight, total length, body length and body height data, and variance analysis and multiple comparisons were performed.

Thirteen fish were randomly selected in equal amounts from SD and TJ, and 14 were selected from QHD. The fin tissue of these 40 fish was collected, and DNA was extracted for sequencing.

### 2.3. DNA Isolation and Sequencing

Genomic DNA was extracted using a TIANamp marine animal DNA Extraction kit (TIANGEN, Beijing, China) according to the kit protocol. DNA quality was checked by 1% agarose gel electrophoresis and a NanoDrop 2000 (Thermo Fisher Scientific, Waltham, MA, USA) spectrophotometer. Qualified DNA samples were randomly interrupted by a Covaris crusher (Covaris, Woburn, MA, USA) with a length of 350 bp. A TruSeq Library Construction Kit (Illumina, San Diego, CA, USA) was used for library construction according to the manufacturer’s instructions. End repair, polyadenylation, the addition of a sequencing adapter, purification, and PCR amplification were conducted for the library preparation of DNA fragments. Illumina NGS technology was used to sequence the constructed libraries. After the library was constructed, Qubit3.0 was used for preliminary quantification, diluted to 1 ng/μL; then, an Agilent 2100 was used to detect the insert size of the library. After the insert size met the requirements, the effective concentration of the library was accurately quantified by qPCR (effective concentration of the library > 2 nM) to ensure the quality of the library. After the library inspection was qualified, PE150 paired-end sequencing was performed on the Illumina platform according to the effective concentration of the library and data output requirements.

### 2.4. Variant Discovery and Quality Control

The raw data obtained through sequencing were filtered to remove low-quality sequencing data using fastp v0.20.0 software [21]. The filtering standards were as follows: (1) Reads containing Adapter sequences were filtered. (2) Reads with more than 10% of the total number of bases in a single end read that could not be determined were filtered. (3) Reads with low-quality bases exceeding 50% of the read length were filtered. Effective, high-quality sequencing data were aligned to the reference genome by BWA (--mem) [22] software, and the alignment results were used to remove duplicates (rmdup parameter) with SAMTOOLS [23]. SAMTOOLS, BCFTOOLS (filter -e ‘QUAL < 10 || DP < 2’) [24] software was used for population SNP detection. The polymorphism sites in the population were detected using the Eyes model, and high-quality SNPs were obtained by filtering and screening as follows: (1) SNPs with mass values below Q20 were filtered out. (2) SNPs < 5 bp apart were filtered out. (3) SNPS with a coverage depth below 2 times the average were filtered out. Finally, SnpEff [25] software was used to apply gene-based annotation, region-based annotation, filter-based annotation, and other functionalities to SNP detection results.

### 2.5. Population Genetic Diversity Analysis

Based on the filtered SNP variation file, PLINK (--het; --hardy) software [26] calculated the observed heterozygosity (Ho) and expected heterozygosity (He) of different population variation sites. The population fixation index (Fst) and nucleotide diversity (θπ) were calculated using VCFtools (--window-pi 50,000 --window-pi-step 10,000; --fst-window-size 50,000 --fst-window-step 10,000) [27], with the sliding window size set to 50 Kb and the window step set to 10 Kb. The strong selection signal within the selected region between the groups was obtained and visualized by drawing scatter plots in R. The genome-wide heterozygous rate (number of heterozygous SNPs/genome size) was calculated.

### 2.6. Linkage Disequilibrium (LD) and Population Structure Analysis

The correlation coefficient (r^2^) measures the degree of LD between different SNPs. The observed pair-wise LD (r^2^) was calculated using PopLDdecay [28] software (-MaxDist 500). The population structure was studied using principal component analysis (PCA) and population structure analysis of three populations. Using PLINK (v1.90) (--pca 20) software, individuals were clustered into subpopulations according to principal components based on the degree of difference in SNPs, and redundant SNPs with strong linkage were removed. The PLINK -indep-pairwise parameter was set to 100 20 0.5. Finally, individual ancestries were estimated using the model-based maximum likelihood method implemented in ADMIXTURE [29], which utilizes a fast numerical optimization algorithm to calculate estimates. Assuming that the population is composed of several subgroups (k = x), the simulation algorithm was used to find the most reasonable sample classification method in the case of k = x. Then, according to the maximum likelihood value of each simulation, we identified the k value that is most suitable for this population.

### 2.7. Screening for Selective Sweeps

The Fst and π ratio method was used to analyze the selection signals of different populations in pairs, and the intersection extracted from the value range of the top 5% was a significant selected region. The genes contained in these regions were the selected genes. The selected region was extracted using BEDtools software and genetically annotated based on the annotation GFF file.

### 2.8. KEGG Enrichment Analysis of Candidate Genes

KofamKOALA [30] software was used to perform KEGG enrichment analysis of *T. fasciatus* genes based on the HMM model, and a database for KEGG enrichment analysis was constructed. The clusterprofiler [31] R package was used for KEGG analysis of the selected genes. The resulting *p* value was corrected using the Benjamini–Hochberg (BH) method to obtain the adjusted *p* value (P_adj_), and a P_adj_ value < 0.05 was considered statistically significant.

## 3. Results

### 3.1. Comparison of Growth Traits in Different Geographic Groups

The differences in weight, body length, body height, and total length indicators between different populations are shown in Figure 2. The weight, body length, body height, and total length of individuals in the TJ population were significantly higher than in the QHD and SD populations. The body length, height, and total length of individuals in the QHD population were significantly higher than in the SD population, but there was no significant difference in weight between the QHD and SD populations. The biometric data (minimum, maximum, average and standard error) of different populations of roughskin sculpin are shown in the Appendix A. Interestingly, we found that compared with the wild population in QHD, the SD population, as an artificial breeding population, experienced no increase in morphometric parameters. Overall, the morphometric parameters of the TJ population, which was also subject to artificial breeding, was significantly superior to those of the QHD population.

### 3.2. SNP Characteristics

All 40 fishes were genotyped for 2,610,157 high-quality SNPs and 494,698 InDels in this study. The variants were filtered with a call rate > 90% and MAF > 0.01 in each population. SnpEff annotation revealed that 52.8% of variants were located in intergenic regions, and 47.2% were located in gene regions, including exons, introns, and untranslated regions. The variants on introns accounted predominantly for variants located in the gene region (41.4%). Only 6.0% of variants were located in the exon region, containing 152,974 synonymous and 276,614 nonsynonymous substitutions (Table 1).

### 3.3. Genetic Diversity

Genetic diversity among populations was assessed by estimating observed and expected heterozygosity, genetic distance, and nucleotide diversity, as shown in Table 2. The observed heterozygosity (0.3524~0.3549) and expected heterozygosity (0.3343~0.3346) of the three groups did not differ significantly. The observed heterozygosity was not significantly different from the expected heterozygosity. The proportions of the polymorphic SNPs were 98.39%, 98.89%, and 98.89% in the wild and two farmed populations, respectively. The proportions of the polymorphic SNPs in the two farmed populations were similar but higher than in the wild population. Genetic distance (D) was used to measure the degree of genetic difference among samples within a population. The approximate genetic distance for the QHD, SD, and TJ was 0.274212, 0.278117, and 0.277829, respectively. 

### 3.4. Population Differentiation and Linkage Disequilibrium

By analyzing the degree of genetic differentiation among the three populations, it was found that the QHD population exhibited the highest genetic differentiation from the SD and TJ populations, with a mean Fst of 0.02. The SD population had the lowest genetic differentiation compared with the TJ population, with a mean Fst of 0.005 (Figure 3). Because the Fst values of the three populations were all less than 0.05, we considered that the genetic composition of the three populations was similar and that the degree of genetic differentiation was low. Genome-wide linkage imbalance (LD) analysis showed that the linkage degree of SPNs in the three groups was low (0~0.15) (Figure 4). The decay of the LD coefficient (r^2^) was slower in the QHD population than in the SD and TJ populations, which were cultured and subjected to a certain degree of selection. In addition, we found that the r^2^ attenuation decay curves of the SD and TJ populations overlapped, indicating that the two populations exhibited a low degree of differentiation, consistent with the Fst results.

Principal component analyses were performed on the QHD, SD, and TJ populations to explore their population structures. The first three were selected as clustering factors according to the variance-explained rate of principal component factors. The QHD population could be clearly distinguished from the SD and TJ populations by three principal components (Figure 5). The SD population overlapped with the TJ population. During the ADMIXTURE analysis, none of the k values’ cross-validations (CVs) converged. In this study, we also determined the optimal number of subgroups (k = 3) by taking the actual number of geographical populations and dividing those three populations into two (Figure 6). The SD and TJ populations were clustered into one population, consistent with the Fst and principal component analysis results.

### 3.5. Selective Sweep and Selected Gene Annotation

Combined with Fst and the ratio of PI between different populations, strong selection signals were detected, and target genes were screened. During the comparisons of QHD and SD, QHD and TJ, and SD and TJ populations, we took the highest 5% of Fst and the window with the highest 5% (high top5%) and the lowest 5% (low top5%) Pi ratios as potential selection regions and merged adjacent and overlapping windows (Figure 7A, Figure 8A and Figure 9A). We used QHD as the reference population, since it is a wild population. At a significance level of 5%, 709 selective sweep regions were found in the TJ population compared to the QHD population. Approximately 35.45 MB and 523 genes were detected in the selective region. Furthermore, 629 selective sweep regions were found in the SD population compared to the QHD population, and approximately 31.45 MB and 529 genes were detected in the selective region. A previous population structure analysis showed that the QHD and TJ populations were closely related. Accordingly, the selected genes of the two groups were compared. It was found that 44.93% of genes between the two populations were homologous. Compared to SD, the TJ population had 457 selected regions containing 413 genes.

The annotated genes underwent KEGG enrichment analysis to narrow the selected gene range. A total of 300 of the 500 genes in the TJ population were annotated to the corresponding KOs, and enrichment analysis showed that these genes were enriched in apoptosis, phenylalanine metabolism, and other pathways, but no significant functional clusters were identified during KEGG pathway enrichment analysis (Figure 7B). The enrichment analysis revealed several genes linked to the autophagy-animal, toll-like receptor signaling pathway, phenylalanine metabolism, ECM–receptor interaction, the Wnt signaling pathway, and other pathways. Enrichment analysis of selected genes in the SD population showed similar enrichment results to the TJ population, regardless of the significance level or the enrichment pathway. Besides pathways enriched in the TJ population, other pathways enriched in the SD population included cell adhesion molecules (Figure 8B). KEGG enrichment analysis of selected genes in the TJ population compared to the SD population showed that the results differed from those with the QHD population as the reference population. The selected genes were associated with other pathways, such as cytokine–cytokine receptor interaction, cytokine receptors, and PPAR signaling pathways, although no statistical significance was found (Figure 9B).

## 4. Discussion

The roughskin sculpin (*T. fasciatus*) was once widely distributed in the Northwest Pacific, but its habitat has been fragmented due to human activities such as dam construction and environmental pollution, resulting in a significant decrease in its population. Nowadays, it is only found in certain river basins of the Yellow and Bohai Seas and the East China Sea in China. The endangered status of the roughskin sculpin has prompted the implementation of conservation measures, such as stocking and releasing. Research on artificial breeding technology of the roughskin sculpin, including the production and aquaculture of the species, has made breakthroughs that not only protect the population of the roughskin sculpin but also meet market demand. Due to differences in breeding methods, growth patterns, feed sources, and feeding frequencies, there are differences in growth characteristics between wild and cultured populations [32,33]. In the natural sea area, roughskin sculpin mainly eat small shrimps and small fish, but most of Songjiang bass eat small shrimps. *Palaemonetes sinensis sollaud* and *Caridina nilotica gracilipes de Man* are the main shrimps ingested by roughskin sculpin. The breeding population of roughskin sculpin is fed commercial compound feed. The main source of protein in commercial feed is fish meal. Given that the cultured populations are fed on a fixed basis and have access to sufficient food during the breeding process, the morphological characteristics of the cultured population are significantly better than those of the wild population. Therefore, morphological characteristics represent important and stable indicators for distinguishing sea bream and sea bass from other marine fish [34,35,36]. In this study, the two cultured populations showed different morphological characteristics than wild populations. The three populations were simultaneously assessed to avoid the effect of confounders such as growth time. The weighted index of the SD population was similar to that of the QHD wild population, but the length, height, and other indices were significantly lower than those of the QHD population, which may be attributed to the fact that the artificial synthetic feed provided to the SD population could not meet the nutritional needs of the roughskin sculpin. Accordingly, the morphological characteristics of the TJ population, which is also a cultured population, were significantly better than those of the wild population. Moreover, it should be borne in mind that the survival environment leads to differences in body shape. Wild roughskin sculpin may require a longer body shape to adapt to the natural survival environment.

Population genetic diversity refers to species’ genetic diversity within an area. Higher genetic diversity means that a species is more resilient to its environment. Current evidence suggests that changes in population size, structure, genetic mutations, and other factors can affect the population’s genetic diversity. A study by Dewoody and Avise showed that the average heterozygosity of marine fish microsatellites was 0.79 and that the average heterozygosity of anadromous fish was 0.68 [37]. Li et al. revealed that the average observed heterozygosity of microsatellites in the roughskin sculpin population (0.83) was higher than the average of marine fish and higher than other Catadromous migration fish, such as the Japanese eel and the European eel [12,38,39]. Wild marine fish species *Pangasianodon Hypophthalmus* and *Oreochromis niloticus* have previously shown genome-wide SNP heterozygosity of 0.04–0.60. In this study, the SNP heterozygosity of the wild population was found to be 0.335, which is comparable to that of the wild *O. niloticus* population [40,41]. However, Li et al. showed that the average heterozygosity of SNP in wild roughskin sculpin was 0.121, unlike the results of this study [7]. We speculate that this is due to the number of SNPs used for analysis—2,610,157 SNPs were used in this study to calculate heterozygosity, which is higher than that reported in Li et al.’s study (*n* = 10,153). In this respect, the population heterozygosity calculated by microsatellites was higher than that calculated by SNPs, suggesting that the number of markers affects heterozygosity. Theoretically, when the species range is limited and its effective population is decreased, its genetic diversity generally decreases due to genetic drift and inbreeding decline [42]. However, in this study, roughskin sculpin maintained a high level of genetic diversity, possibly due to its population size, which mitigated the effect of genetic drift. Higher levels of genetic diversity have also been found in populations of some endangered species [43,44,45]. Affected by genetic drift and inbreeding, the genetic diversity of farmed populations was lower than that of wild populations. In the present study, we found that the heterozygosity of farmed and wild roughskin sculpin was similar. The roughskin sculpin is a fish with a life cycle of only one year, and both male and female fish die after spawning [46]. Therefore, the age structure of the roughskin sculpin population in the natural environment is relatively simple, avoiding an excessively close relationship between the collected samples caused by differences in generations. For the farmed population, the eggs of roughskin sculpin were bred by catching wild roughskin sculpin as parents; each broodstock can only be used for one year, which indirectly slows down the phenomenon of source degradation of the breeding population [47]. Overall, the above reasons account for the absence of significant differences among genetic diversity indicators observed in this study, such as heterozygosity, the proportions of the polymorphic SNPs, and genetic distance between wild and farmed roughskin sculpin populations. The main goals of conservation genetics is to protect the genetic diversity of species, improve the environmental adaptability of species, and reduce the risk of extinction. According to the results of this study, the genetic diversity of the wild population and that of the cultured population of roughskin sculpin are similar. In other words, it is feasible to restore the population size of roughskin sculpin in the natural sea area by supplementing with artificially bred roughskin sculpin. Moreover, the detection of genetic diversity of cultured populations before release can be used as a standard for stocking. The genetic diversity of roughskin sculpin in the natural sea area was detected regularly, and the effect of stocking was evaluated to a certain extent.

Population structure analysis results based on the genome-wide SNPs showed the genetic structure between the populations of roughskin sculpin. Principal component analysis showed that the three geographical groups of roughskin sculpin could be divided into two genetic groups: the SD and TJ population vs. the QHD population. Population structure analysis further verified the results of the principal component analysis. Although the cross-validation values of the predicted optimal population number did not converge, at K = 3, most individuals in the SD population and the TJ population were divided into one group, and the QHD population was independent of the other group. The SD population sampling point was far away from the TJ population. Given that there is theoretically no large-scale gene exchange, we speculate that the two groups come from the same spawning ground. Many molecular markers can be obtained by genome-wide testing, and due to their wide distribution in the genome, the fine genetic structure differences between populations can be detected. Compared with the QHD population, the differentiation coefficients of the SD and TJ populations were 0.0156 and 0.0158, respectively, with weak genetic differentiation. However, the SD and TJ populations did not show apparent genetic differentiation (Fst = 0.0039), consistent with the population structure analysis and principal component analysis results. Consistently, in Li’s study, the genetic differentiation of wild roughskin sculpin from different geographical groups was not apparent [7].

Fst and Π are effective methods for detecting selective elimination regions, especially when mining functional areas closely related to the living environment, which can yield strong selection signals [48]. Indeed, the living environment, food source, composition, and other aspects of the breeding population differ completely from those of the wild population. Under the influence of human beings, the breeding population’s relevant traits can be selected to better adapt to the breeding environment. Selective sweep analysis showed that captivity caused significant changes in the genome of roughskin sculpin. The upstream of the gene encoding L-amino acid oxidase (LAAO) was selected by comparing two cultured populations and one wild population. LAAO is an important oxidoreductase involved in the oxidative metabolism of amino acids in organisms. It can catalyze the oxidative deamination of L-amino acids with oxygen molecules as electron acceptors to produce corresponding keto acids, ammonia (NH_3_), and hydrogen peroxide (H_2_O_2_), which involve the metabolic pathways of tryptophan, phenylalanine, and tyrosine. Because the protein source of artificial compound feed is different from that of natural feed, under long-term regular feeding and management, the selected genes related to amino acid metabolism may play an important role in the process of metabolizing artificial compound feed. It has been established that a higher standard metabolic rate can lead to increased efficiency in food intake needed for growth. In this respect, a previous study comparing transcriptomic profiles of farmed and wild Atlantic salmon showed that differentially expressed genes were highly correlated with energy metabolism [49].

## 5. Conclusions

Overall, 2,610,157 high-quality SNPs and 494,698 InDels were identified using whole-genome sequencing, and the genetic diversity and population structure of two farmed and one wild population of roughskin sculpin were determined. The levels of genetic diversity were similar in all populations. Analysis of Fst values indicated low levels of genetic differentiation between the two farmed populations and wild populations and the lowest levels of genetic differentiation between the two farmed populations. According to population structure and principal component analysis, two clusters were detected in all the populations, and the two farmed populations could be clustered into one population, suggesting seedling exchange between the Shandong and Tianjin populations, which might affect the adaptability of roughskin sculpin. Selective sweep analysis showed that the selected genes in the two populations were related to amino acid metabolism, which might be caused by artificial feeding. These results provide essential information for the conservation of genetic diversity in roughskin sculping. Accordingly, corresponding genetic diversity conservation measures can be formulated based on to the genotyping results. Finally, this study demonstrates that whole-genome resequencing is an efficient genome-wide SNP discovery tool that can accurately evaluate population genetics in roughskin sculpin.

## Figures and Tables

**Figure 1 biology-12-01427-f001:**
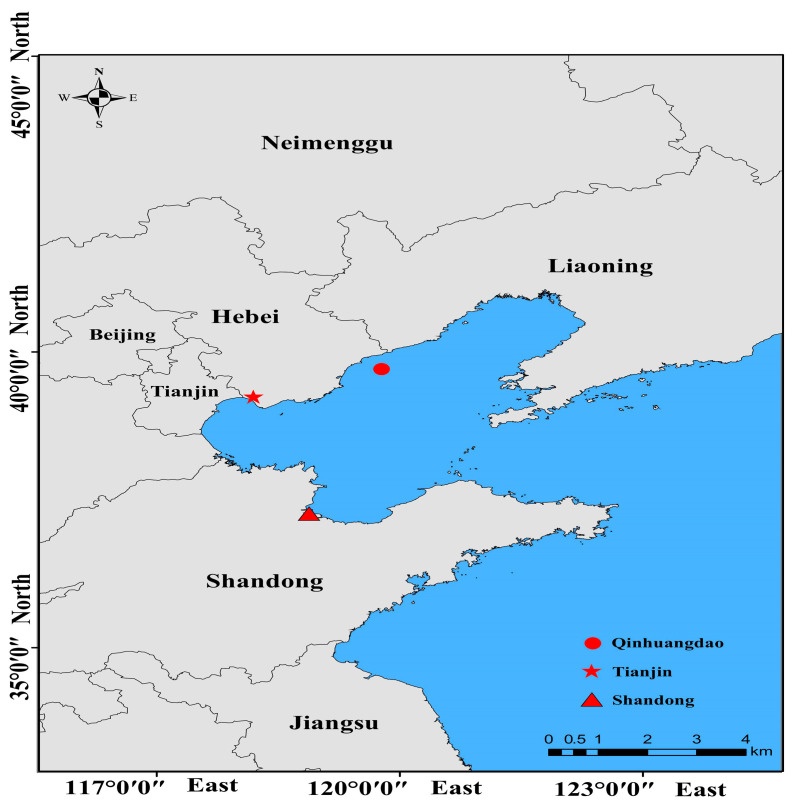
Map of sampling sites for roughskin sculpin.

**Figure 2 biology-12-01427-f002:**
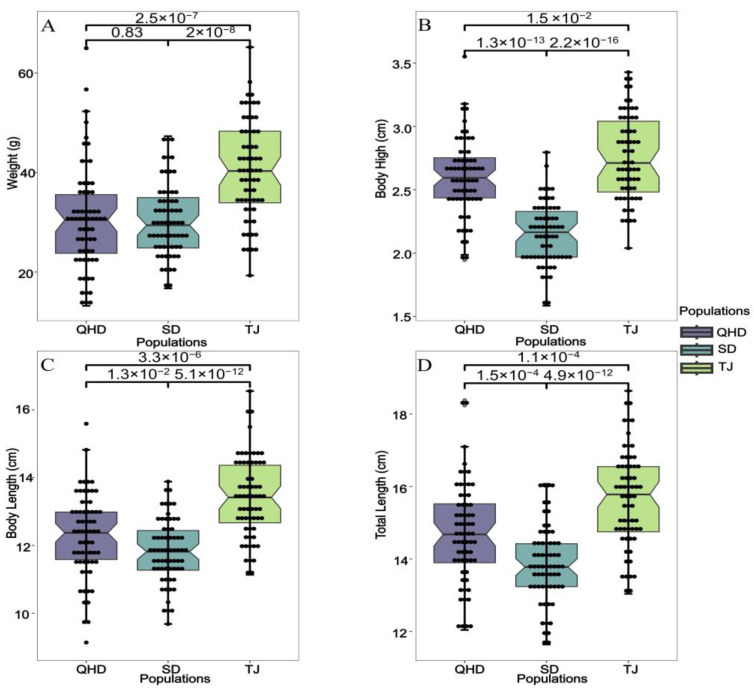
Box plot of morphometric statistics for growth indices of different populations of roughskin sculpin. The number of samples measured in each population was 64. (**A**) Weight; (**B**) body height; (**C**) body length; (**D**) total length.

**Figure 3 biology-12-01427-f003:**
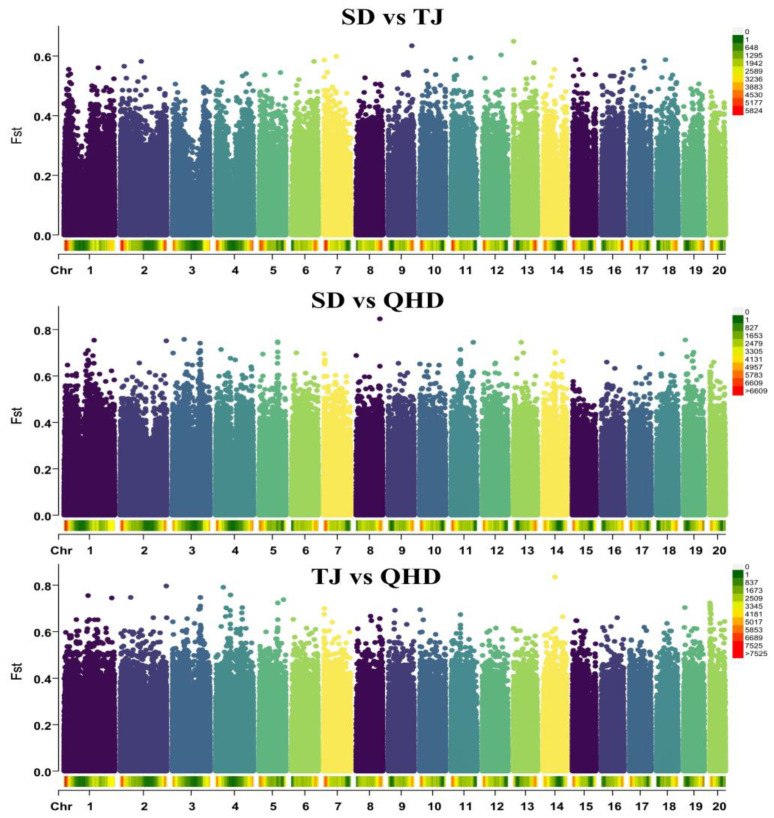
Manhattan plots of Fst values. The X axis represents different chromosome names, and the Y axis represents the Fst values within the corresponding chromosome window. The different colors of point represent different chromosomes.

**Figure 4 biology-12-01427-f004:**
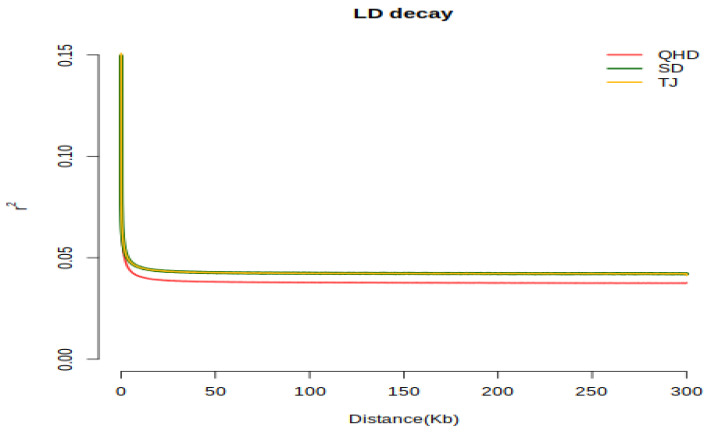
Genome-wide linkage disequilibrium (LD) decay plots. The x axis represents the distance of LD, and the y axis represents the linkage disequilibrium correlation coefficient.

**Figure 5 biology-12-01427-f005:**
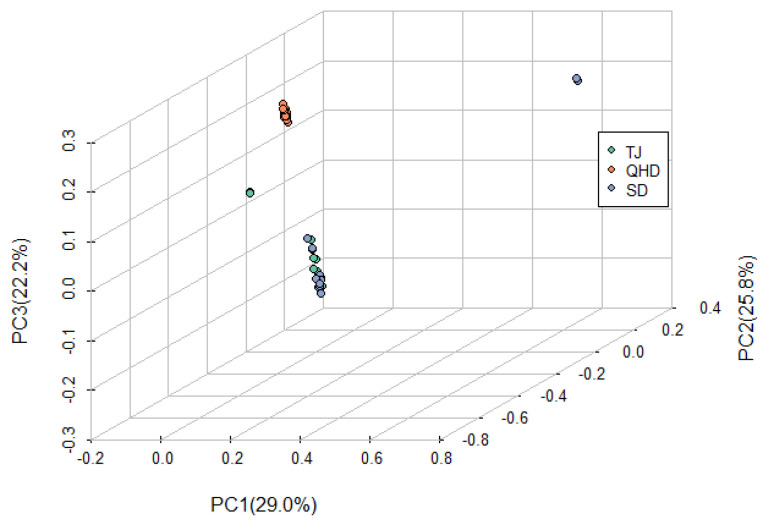
Sample structure indicated by principal component analysis (PCA) with the first three principal components.

**Figure 6 biology-12-01427-f006:**
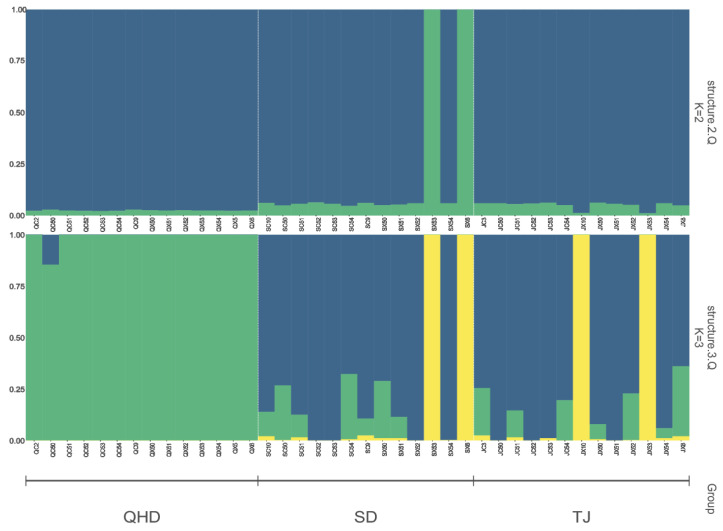
Admixture structure plot. Each column represents an individual, where the length of the different colored segments indicates the proportion of the individual that belongs to different groups.

**Figure 7 biology-12-01427-f007:**
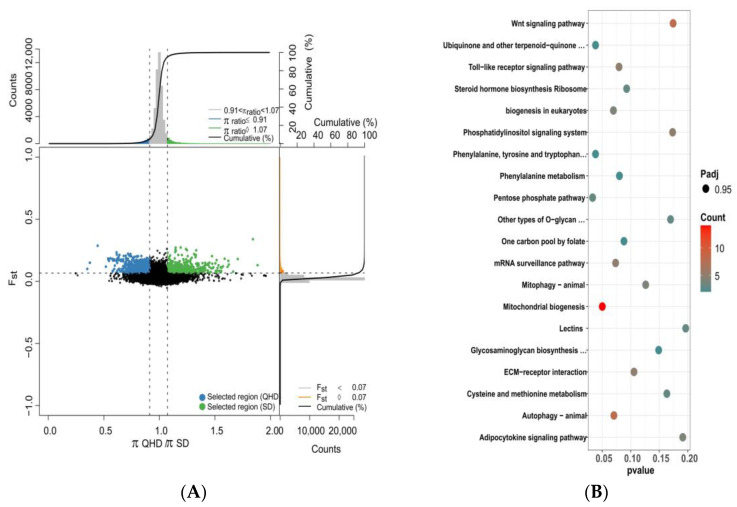
Identification of the domestication-selective sweeps and the KEGG enrichment analysis. (**A**) The θπ ratio and Fst values of QHD and SD (green points indicate selection; black points represent not selection.). Line represents the selection threshold of 0.05. (**B**) The KEGG enrichment analysis of selected genes of SD.

**Figure 8 biology-12-01427-f008:**
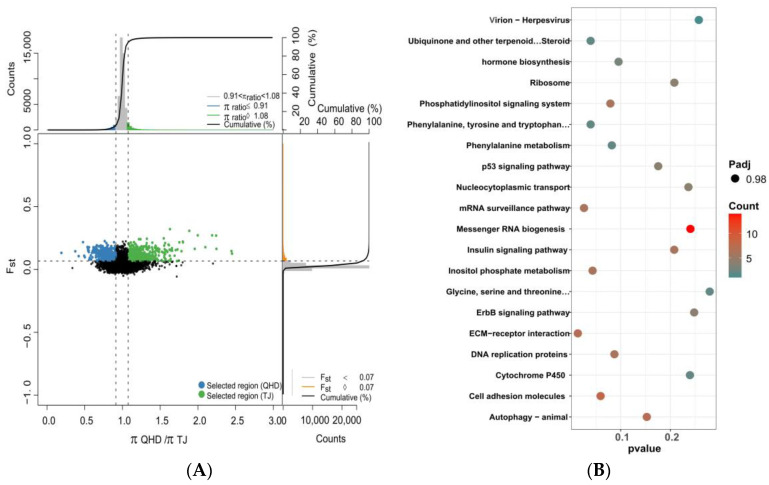
Identification of the domestication-selective sweeps and the KEGG enrichment analysis. (**A**) The θπ ratio and Fst values of QHD and TJ (green points indicate selection; black points represent not selection.). Line represents the selection threshold of 0.05. (**B**) The KEGG enrichment analysis of selected genes of TJ.

**Figure 9 biology-12-01427-f009:**
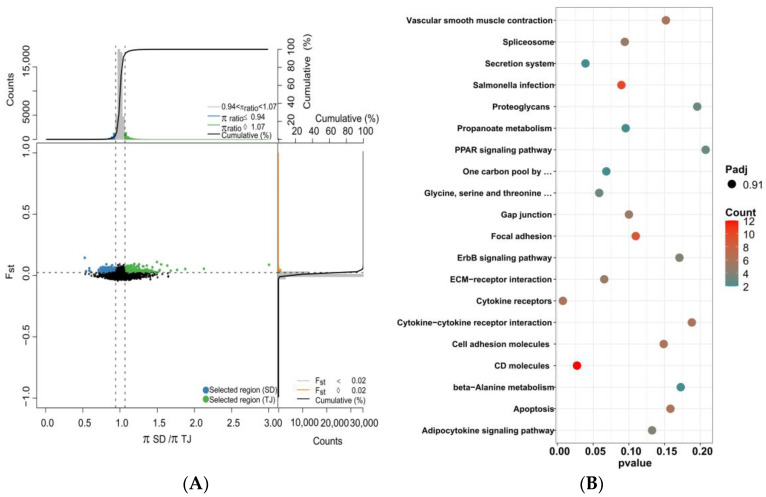
Identification of the domestication-selective sweeps and the KEGG enrichment analysis. (**A**) The θπ ratio and Fst values of SD and TJ (green points indicate selection; black points represent not selection.). Line represents the selection threshold of 0.05. (**B**) The KEGG enrichment analysis of selected genes of TJ.

**Table 1 biology-12-01427-t001:** The number of variant effects by region and type.

Category	Type	Count	Percent
Region	Intergenic	3,331,449	47.0%
	Intron	2,985,075	41.4%
	Exon	429,588	6.0%
	5′ UTR	125,011	3.9%
	3′ UTR	280,506	1.7%
Mutation	Synonymous	152,974	2.1%
	Non-synonymous	276,614	3.9%

**Table 2 biology-12-01427-t002:** Summary of genetic diversity measurement for three populations.

Statistic	QHD	SD	TJ
Ho	0.3524	0.3539	0.3549
He	0.3344	0.3343	0.3346
MAF	0.2437	0.2432	0.2434
P_N_	98.39%	98.89%	98.89%
D	0.274212	0.278117	0.277829

## Data Availability

Raw DNA sequencing reads were deposited in the NCBI under project accession No. PRJNA945830.

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
