# Peer review of "Genetic Diversity and Signatures of Selection in the Roughskin Sculpin (Trachidermus fasciatus) Revealed by Whole Genome Sequencing"

_biology, 2023, doi:10.3390/biology12111427_

Round 1
Reviewer 1 Report
Comments and Suggestions for Authors
Clearly state the specific objectives of your study in the introduction. Additionally, provide more context regarding the significance of investigating genetic diversity and selection in Trachidermus fasciatus to set the stage for your readers.
Include more details on the sampling strategy, such as geographic locations and sample sizes. This information is essential for readers to assess the representativeness of your study population.
Clarify the bioinformatics tools and parameters used for data analysis. This will ensure reproducibility and allow readers to assess the robustness of your findings.
Discuss the implications of your findings in the broader context of population genetics and conservation. Address how the observed genetic diversity and selection signatures align with the species' ecological and evolutionary history.
Compare your results with existing literature, highlighting similarities and differences. This will help readers contextualize your findings within the current body of knowledge.
Elaborate on the specific genomic regions identified as under selection. Discuss potential functional implications and ecological relevance. Consider citing relevant studies that support or contrast your findings.
Comments on the Quality of English LanguageMinor editing of English language required
Reviewer 2 Report
Comments and Suggestions for Authors
The decision on the manuscript with ID (biology-2655283) is “Minor Revisions”. The authors should prepare a point-by-point response to the comments raised by the anonymous reviewer. The authors should follow the comments presented in the PDF file. I have several questions.
Q1. Line 122: Which fin that you choose? And what is the cause of its selection?
Q2. Line 119: Water quality parameters (means ± SE) (temperature, DO, pH, salinity) should be described for each fish group.
Q3. Line 116: Add data on the weights (g; means ± SE) for each sampled fish group.
Q4. For all kits, instruments, and apparatuses used in the manuscript, authors should write the sources in detail (company name, city, country).
Q5. Figures 1-2: Add the number of samples.
Q6. Line 312: Add suitable references.
Q7. Revise Latin names in the reference section.

Author Response
Dear Reviewers:
We sincerely thank the editor and all reviewers for their valuable feedback that we have used to improve the quality of our manuscript. According to your nice suggestions, we have made extensive corrections to our previous draft, the detailed corrections are listed below.
Q1. Line 122: Which fin that you choose? And what is the cause of its selection?
Thank you very much for the reviewer 's question. We chose the ventral fin. Because the ventral fin can extract dna, and cutting off the ventral fin will not affect its survival.。
Q2. Line 119: Water quality parameters (means ± SE) (temperature, DO, pH, salinity) should be described for each fish group.
We agree with the reviewer’s assessment. We supplemented the living environment of different populations at the corresponding locations, including temperature, dissolved oxygen, pH, and salinity. “According to the origin of the population, they were named QHD, TJ, and SD, respectively. The SD (temperature: 12±1℃, DO: 6.5±0.5 mg/L, pH:7.5±0.5, salinity:24±1‰) and TJ groups (temperature: 11±1℃, DO: 6.5±0.5 mg/L, pH:7.5±0.5, salinity:24±1‰) were cultured, and QHD (temperature: 10±1℃, DO: 8.5±0.5 mg/L, pH: 8.0±0.5, salinity:31±1‰) was a wild population.”
Q3. Line 116: Add data on the weights (g; means ± SE) for each sampled fish group.
Thank you for your suggestion. We added the average body weight of different geographic populations.” A total of 192 T. fasciatus were collected from three different geographic populations (Qinhuangdao City of Hebei province (n=64, Weight:30.66±10.4g), Tianjin Municipality (n=64, Weight:40.46±9.84g), and Dongying City of Shandong province (n=64, Weight:30.47±7.21g), China).”
Q4. For all kits, instruments, and apparatuses used in the manuscript, authors should write the sources in detail (company name, city, country).””
Thank you for your suggestion. We added kits, instruments, and apparatuses production companies and countries."Genomic DNA was extracted using the TIANamp marine animal DNA Extraction kit (TIANGEN, Beijing, China) according to the kit protocol. DNA quality was checked by 1% agarose gel electrophoresis and NanoDrop 2000 (Thermo Fisher Scientific, US) spectrophotometer. Qualified DNA samples were randomly interrupted by a Covaris crusher (Covaris, US) with a length of 350 bp. The TruSeq Library Construction Kit (Illumina, US) was used for library construction according to the manufacturer's instructions."
Q5. Figures 1-2: Add the number of samples.
Thank you for your suggestion. We added the number of samples for each group in the figure explanation.
Q6. Line 312: Add suitable references.
As suggested by the reviewer, we have added more references to support this idea. “Yi WG, Liang P, Liang TH, Huang YL. Roughskin Sculpin (Trachidermus fasciatus) in Yalu River Basin. Hebei Fisheries. 2006, 2, 53-54:(in Chinese)”
Q7. Revise Latin names in the reference section.
Thank you very much for the reviewer 's reminder. We have examined the Latin name of the species in the references and modified it to italic.
Reviewer 3 Report
Comments and Suggestions for Authors
Review for the paper “Genetic diversity and signatures of selection in the roughskin sculpin (Trachidermus fasciatus) revealed by whole genome sequencing” by Lize San, Zhongwei He, Yufeng Liu, Yitong Zhang, Wei Cao, Jiangong Ren, Tian Han, Bingbu Li, Guixing Wang, Yufen Wang, Jilun Hou submitted to "Biology".
General comment.
The study aimed to investigate the genetic diversity of Trachidermus fasciatus, a critically endangered fish species of great economic importance in China. Despite efforts in artificial breeding, the genetic diversity of the natural populations of this species remained uncertain, especially after the release of farmed individuals. To address this, the authors conducted whole-genome sequencing of two farmed populations and one wild population of Roughskin sculpin. The farmed populations exhibited consistent levels of genomic polymorphism, but differed in population structure from the wild population, although the genetic differentiation was relatively low. Further analysis revealed dissimilarities in amino acid metabolism, likely influenced by variations in food sources. These findings have significant implications for the conservation of this endangered fish species.
Recommendations:
1) L 116–118. The authors should include a map of the study area, with a coordinate grid, highlighting the three sampling locations. Additionally, they should provide a detailed description of the sampling procedure, including the devices used, the depths sampled, and the season and year of sampling.
2) L 122. The authors should provide further details about the sampling procedures, including information about the instruments used, the euthanasia protocols employed, and the weight of the samples collected.
3) L 192. The authors have made claims regarding significant differences between populations; however, they did not clarify the specific method used to test the data. It is recommended that they update the Materials and Methods section to include this information.
4) L 199. The authors mentioned the "growth index" but did not provide an explanation of how this index was calculated. It is advisable for them to include this information in the Materials and Methods section. Additionally, to enhance comprehension, the authors should consider providing biometric data (such as minimum, maximum, mean, and standard error) in a supplementary table.
5) L 317-320. When discussing the influence of food sources on the morphometric characteristics of Roughskin sculpin, it is crucial for the authors to provide the corresponding data. This would involve informing the reader about the natural diets of this species in their habitats and detailing the diets used in the farms.
6) In the Discussion section, it would be valuable for the authors to explain how their data can be applied in the context of management and conservation practices.
7) Furthermore, it is important to review and align the references with the Instructions for Authors.
Specific remarks.
L 85. Consider replacing “the Atlantic Ocean of eight populations” with “the Atlantic Ocean’s eight populations”
L 95. Consider replacing “Based on the variation information, biological” with “Through the analysis of genetic variations, various biological”
L 120. Consider replacing “Their growth traits” with “Their morphometric parameters”
L 145. Consider replacing “low quality bases” with “low-quality bases”
L 206-212. Consider replacing “varints” with “variants”
L 229. Consider replacing “was the most genetically differentiation” with “exhibited the highest genetic differentiation”
L 262. Consider replacing “belongs to a different groups” with “belongs to different groups”
L 341. Consider replacing “The known wild marine fish Catfish (Pangasianodon Hypophthalmus), and Nile tilapia (Oreochromis niloticus) had genome-wide SNP heterozygosity of 0.04-0.60, while the wild population SNP in this study was 0.335, similar to wild O. niloticus” with “The wild marine fish species Pangasianodon Hypophthalmus and Oreochromis niloticus have previously shown a genome-wide SNP heterozygosity of 0.04-0.60. In this study, the SNP heterozygosity of the wild population was found to be 0.335, which is comparable to that of the wild O. niloticus population.”
L 393. Consider replacing “By compairson” with “By comparison”
Comments on the Quality of English LanguageMinor revision
Author Response
Dear Reviewers:
Thank you very much for your comments and professional advice. These opinions help to improve academic rigor of our article. Based on your suggestion and' request, we have made corrected modifications on the revised manuscript, the detailed corrections are listed below.
.1) L 116–118. The authors should include a map of the study area, with a coordinate grid, highlighting the three sampling locations. Additionally, they should provide a detailed description of the sampling procedure, including the devices used, the depths sampled, and the season and year of sampling
Thank you for your suggestion. We have added detailed information about the sample collection process. “The breeding group was taken from the farm, and the sampling time was November 2022. Wild populations are caught by bottom trawling. The fishing site was in the off-shore area of the Bohai Sea, the fishing depth range was 4~6 m, the fishing route was 30 km, and the fishing time was November 2022.”
2) L 122. The authors should provide further details about the sampling procedures, including information about the instruments used, the euthanasia protocols employed, and the weight of the samples collected.
Thank you for your suggestion. We supplemented the details of the sampling procedure. “Four morphological traits of body weight (BW), body length (BL), body height (BH) and total length (TL) were measured after anesthesia with 30 g/m3 MS-222. The weight was weighed by electronic balance with an accuracy of 0.1 g. The body length and height were measured with a vernier caliper, accurate to 0.1 cm. The measurement methods of the total length, body length and body height of the sculpin are as follows:
Total length: The vertical distance from the front end of the maxillary kiss to the end of the caudal fin. Body length: The vertical distance from the front end of the maxillary kiss to the base of the caudal fin. Body height: The vertical distance at the highest point of the fish body.”
3) L 192. The authors have made claims regarding significant differences between populations; however, they did not clarify the specific method used to test the data. It is recommended that they update the Materials and Methods section to include this information.
Thank you for your suggestion. We supplemented the details of the data processing process to compare the significance of growth traits between different populations. “SPSS 17.0 was used to test the normality of all measured body weight, total length, body length and body height data, and variance analysis and multiple comparisons were performed.”
4) L 199. The authors mentioned the "growth index" but did not provide an explanation of how this index was calculated. It is advisable for them to include this information in the Materials and Methods section. Additionally, to enhance comprehension, the authors should consider providing biometric data (such as minimum, maximum, mean, and standard error) in a supplementary table.
The growth index here refers to the general term for the measured body weight, body length, total length and body height. Maybe the morphometric parameters used here is not accurate, we change it to “morphometric parameters”. We added the measurement data of each group to the supplementary table.
5) L 317-320. When discussing the influence of food sources on the morphometric characteristics of Roughskin sculpin, it is crucial for the authors to provide the corresponding data. This would involve informing the reader about the natural diets of this species in their habitats and detailing the diets used in the farms.
We agree with the reviewer’s assessment. We supplement the details of the natural diet and the diet of the roughskin sculpin in the farm. “In the natural sea area, roughskin sculpin mainly eats small shrimps and also eats small fish, but most of roughskin sculpin eats small shrimps. Palaemonetes sinensis sollaud and Caridina nilotica gracilipes de Man were the main shrimps ingested by roughskin sculpin. The breeding population of roughskin sculpin is fed with commercial compound feed. The main source of protein in commercial feed is fish meal.”
6) In the Discussion section, it would be valuable for the authors to explain how their data can be applied in the context of management and conservation practices.
Thank you for this suggestion. In the discussion section, we supplement our findings for the management and protection value of roughskin sculpin. “The main goal of conservation genetics is to protect the genetic diversity of species, improve the environmental adaptability of species, and reduce the risk of extinction. According to this study results, it was found that the genetic diversity of the wild pop-ulation and the cultured population of the roughskin sculpin was similar. In other words, it is feasible to restore the population size of roughskin sculpin in the natural sea area by putting the artificially bred roughskin. Moreover, the detection of genetic diversity of cultured populations before release can be used as a standard for stocking. The genetic diversity of roughskin sculpin in natural sea area was detected regularly, and the effect of stocking was evaluated to a certain extent.”
7) Furthermore, it is important to review and align the references with the Instructions for Authors.
Thanks to the reviewer 's reminder, we checked the format of the references and confirmed that they met the requirements of the Instructions for Authors.
Specific remarks.
L 85. Consider replacing “the Atlantic Ocean of eight populations” with “the Atlantic Ocean’s eight populations”
Thank you for this suggestion. We have replaced “the Atlantic Ocean of eight populations” with “the Atlantic Ocean’s eight populations”.
L 95. Consider replacing “Based on the variation information, biological” with “Through the analysis of genetic variations, various biological”
Thank you for this suggestion. We have replaced “Based on the variation information, biological” with “Through the analysis of genetic variations, various biological”.
L 120. Consider replacing “Their growth traits” with “Their morphometric parameters”
Thank you for this suggestion. We have replaced “Their growth traits” with “Their morphometric parameters”.
L 145. Consider replacing “low quality bases” with “low-quality bases”
Thank you for this suggestion. We have replaced “low quality bases” with “low-quality bases”.
L 206-212. Consider replacing “varints” with “variants”
Thank you for this suggestion. We have replaced “varints” with “variants”.
L 229. Consider replacing “was the most genetically differentiation” with “exhibited the highest genetic differentiation”
Thank you for this suggestion. We have replaced “was the most genetically differentiation” with “exhibited the highest genetic differentiation”.
L 262. Consider replacing “belongs to a different groups” with “belongs to different groups”
Thank you for this suggestion. We have replaced “belongs to a different groups” with “belongs to different groups”.
L 341. Consider replacing “The known wild marine fish Catfish (Pangasianodon Hypophthalmus), and Nile tilapia (Oreochromis niloticus) had genome-wide SNP heterozygosity of 0.04-0.60, while the wild population SNP in this study was 0.335, similar to wild O. niloticus” with “The wild marine fish species Pangasianodon Hypophthalmus and Oreochromis niloticus have previously shown a genome-wide SNP heterozygosity of 0.04-0.60. In this study, the SNP heterozygosity of the wild population was found to be 0.335, which is comparable to that of the wild O. niloticus population.”
Thank you for this suggestion. We have replaced “The known wild marine fish Catfish (Pangasianodon Hypophthalmus), and Nile tilapia (Oreochromis niloticus) had genome-wide SNP heterozygosity of 0.04-0.60, while the wild population SNP in this study was 0.335, similar to wild O. niloticus” with “The wild marine fish species Pangasianodon Hypophthalmus and Oreochromis niloticus have previously shown a genome-wide SNP heterozygosity of 0.04-0.60. In this study, the SNP heterozygosity of the wild population was found to be 0.335, which is comparable to that of the wild O. niloticus population.”
L 393. Consider replacing “By compairson” with “By comparison”
Thank you for this suggestion. We have replaced “By compairson” with “By comparison”.
We appreciate for Reviewers’ warm work earnestly, and hope the correction will meet with approval. Once again, thank you very much for your comments and suggestions.